# SciPIP: An LLM-based Scientific Paper Idea Proposer

## Abstract

The exponential growth of knowledge and the increasing complexity of interdisciplinary research pose significant challenges for researchers, including information overload and difficulties in exploring novel ideas. The advancements in large language models (LLMs), such as GPT-4, have shown great potential in enhancing idea proposals, but how to effectively utilize large models for reasonable idea proposal has not been thoroughly explored. This paper proposes a scientific paper idea proposer (SciPIP). Based on a user-provided research background, SciPIP retrieves helpful papers from a literature database while leveraging the capabilities of LLMs to generate more novel and feasible ideas. To this end, **1)** we construct a literature retrieval database, extracting lots of papers' multi-dimension information for fast access. Then, a literature retrieval method based on semantics, entity, and citation co-occurrences is proposed to search relevant literature from multiple aspects based on the user-provided background. **2)** After literature retrieval, we introduce dual-path idea proposal strategies, where one path infers solutions from the retrieved literature and the other path generates original ideas through model brainstorming. We then combine the two to achieve a good balance between feasibility and originality. Through extensive experiments on the natural language processing (NLP) field, we demonstrate that SciPIP can retrieve citations similar to those of existing top conference papers and generate many ideas consistent with them. Additionally, we evaluate the originality of other ideas generated by SciPIP using large language models, further validating the effectiveness of our proposed method[1].

## 1 Introduction

With the exponential growth of knowledge and the increasing complexity of interdisciplinary research, machine learning researchers face significant challenges, including information overload and difficulties in exploring novel ideas. Against this backdrop, generating new ideas and innovative concepts efficiently has become a pressing need. Recent advancements in large language models (e.g., GPT-4 (Ouyang et al., 2022), LLaMA (Touvron et al., 2023a;b), Qwen (Bai et al., 2023; Yang et al., 2024), GLM-4 (Zeng et al., 2024), and *etc*), have demonstrated immense potential in enhancing innovation generation. These models are not only capable of understanding and generating complex academic content but also excel in aligning multimodal information, constructing implicit chains of thought, and uncovering non-obvious connections. Leveraging LLMs to assist researchers in generating new ideas holds significant implications for improving research productivity and offers a theoretical foundation and practical guidance for the design of future intelligent research assistants.

Large language model (LLM)-based idea proposers should have the ability to understand the user-provided research background, autonomously retrieve relevant literature, and generate novel and feasible ideas aimed at addressing problems within the given background. Some previous works have proposed their methods (Wang et al., 2024; Baek et al., 2024; Lu et al., 2024). However, existing LLM-based idea proposers still face two challenges: 1) Similar to human researchers, literature retrieval is essential to inspire new ideas and avoid repetitive ideas. Nevertheless, online literature

---

[1]The code and the database will be available soon.

searches are limited to simple keyword matching and cannot fully leverage the user-provided information or the existing literature, leading to incomplete and inaccurate retrieval results. 2) Scientific paper ideas require both novelty and feasibility. However, it is still under-explored about how to enable LLMs to generate entirely new ideas while ensuring their feasibility.

To address the above challenges, we propose our Scientific Paper Idea Proposer (SciPIP). In terms of challenge 1), SciPIP first **constructs a literature retrieval database**. Specifically, we collect a large body of literature from the natural language processing (NLP) field and extract multiple dimensions of information for each paper using techniques such as entity extraction, semantic encoding, summarization, and citation analysis. The information is stored in the database, enabling rapid access to various aspects of the literature during retrieval. Building on this database, we **propose a literature retrieval method** based on semantics, entities, and citation co-occurrence (SEC-based retrieval). In this framework, "semantics" captures the global information of a paper, "entities" focus on local details, and "citation co-occurrence" reflects the hidden relationships uncovered by previous researchers. By matching at these three different levels of granularity, SciPIP offers more comprehensive literature retrieval.

To address the challenge 2), SciPIP **introduces a new method for idea proposal**. It first organizes the retrieved literature and generates ideas inspired by the retrieved works. Subsequently, SciPIP uses a brainstorming approach to generate new ideas without reference to the literature. Depending on the combination of literature-based and brainstorming-based idea generation, we derive three variants of SciPIP. The ideas generated by our method are further filtered and refined to enhance both their novelty and feasibility.

Extensive experiments are conducted to evaluate both idea proposal and literature retrieval on the NLP field. In the retrospective experiments, we use the backgrounds of ACL 2024 papers as inputs to test whether the models could generate the same ideas as those in the published papers, or whether SciPIP could retrieve the same references as the actual citations. Additionally, we conduct innovation experiments, in which the models are prompted to freely propose ideas based on a given background, and the quality of the proposed ideas are assessed by an LLM in terms of novelty, feasibility, and *etc*. The experimental results demonstrate that, compared to existing methods, SciPIP can match more existing ideas and generate ideas with significantly greater novelty and potential.

## 2 RELATED WORKS

Around 60 years ago, scientists began exploring scientific discoveries based on literature retrieval, known as Literature-Based Discovery (LBD) (Swanson, 1986). This approach concentrated on a specific, narrow type of hypothesis: the connections between pairs of concepts, often involving drugs and diseases. LBD introduced the "ABC" model, positing that two concepts A and C are hypothesized to be linked if they appear in conjunction with an intermediate concept B in the literature.

The advent of large language models (LLMs) has revolutionized various fields, and one of the most intriguing applications is their ability to generate scientific hypotheses (Wang et al., 2024; Baek et al., 2024; Lu et al., 2024). LLMs, trained on extensive datasets encompassing a vast array of scientific literature, possess an impressive capacity to recognize patterns and synthesize information across disciplines. By leveraging their advanced natural language processing (NLP) capabilities, these models can propose novel hypotheses that might not be immediately apparent to researchers. The process begins with the model receiving a prompt, typically related to a specific scientific domain, which guides it to generate hypotheses grounded in existing knowledge while also incorporating innovative perspectives. For example, SCIMON (Wang et al., 2024) uses retrieval of "inspirations" from past scientific papers to generate ideas. It explicitly optimizes for novelty by iteratively comparing generated ideas with prior papers and updating them until sufficient novelty is achieved. In contrast, Research Agent (Baek et al., 2024) starts with a core paper as the primary focus and expands its knowledge by connecting information over an academic graph and retrieving entities from an entity-centric knowledge store based on their underlying concepts. It also leverages multiple Reviewing Agents to provide iterative reviews and feedback for refining the generated ideas. AI Scientist leverages large language models (LLMs) to autonomously generate research ideas, implement and execute experiments, search for related works, and produce comprehensive research papers in machine learning. The AI Scientist is designed to automate the entire scientific process, from ideation to experimentation and iterative refinement.

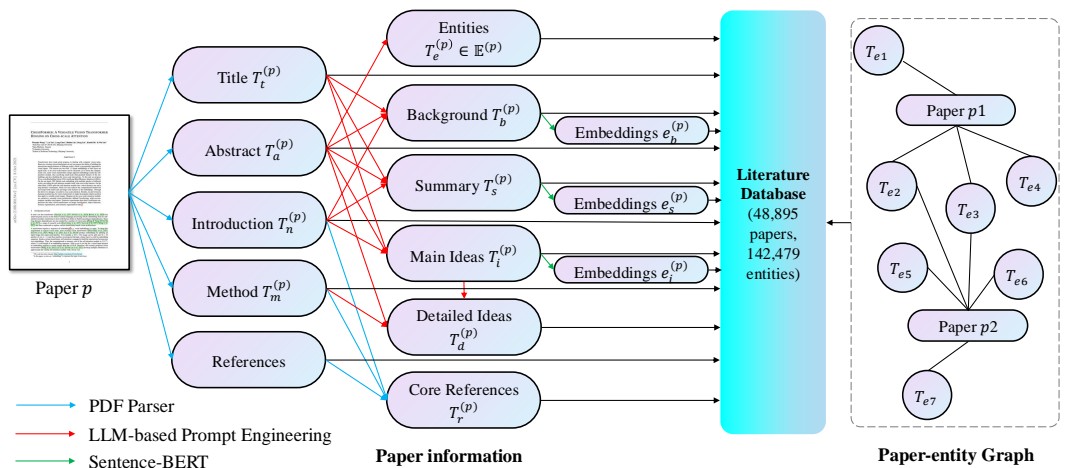

Figure 1: The pipeline of constructing the literature database.

# 3 METHODS

We propose a Scientific Paper Idea Proposer (SciPIP) that takes the user-provided background of a specific research field as input, retrieves relevant literature from the database, and generates novel and feasible ideas. To achieve this, we will first construct a literature database in Section 3.1 for literature retrieval during the idea proposal process. Then, in Section 3.2, we detail how to retrieve literature related to the user-provided background. Finally, in Section 3.3, we outline the process of idea proposal.

## 3.1 LITERATURE DATABASE CONSTRUCTION

Just like human researchers, reading other literature and drawing inspirations from them is an important process for LLMs to generate valuable ideas. However, online literature reading is a very time-consuming process, so we collect a literature database in advance for the following literature retrieval and idea proposal process.

To be specific, we collect papers published in ICLR, NeurIPS, ICML, ACL, NAACL, and EMNLP in past ten years, yielding a database with 48,895 papers. For each paper, we parse the PDF file and extract its title, abstract, introduction, method, and references sections. Then, as shown in Figure 1, given an LLM $f$, we prompt it to read and summarize the paper as:

$$\mathbb{E}^{(p)} = f(\tau_1, T_a^{(p)}),$$
$$(T_b^{(p)}, T_s^{(p)}, T_i^{(p)}) = f(\tau_2, T_t^{(p)}, T_a^{(p)}, T_n^{(p)}), \quad (1)$$
$$T_d^{(p)} = f(\tau_3, T_m^{(p)}, T_i^{(p)}),$$

where $T_t^{(p)}, T_a^{(p)}, T_n^{(p)}, T_m^{(p)}$ are the paper $p$'s title, abstract, introduction, and method sections. $\mathbb{E}^{(p)}, T_b^{(p)}, T_s^{(p)}, T_i^{(p)}, T_d^{(p)}, T_r^{(p)}$ are extracted entities, background, summary, main ideas, detailed ideas, and core references, as shwon in Figure 1. $\tau_i, i \in \{1, 2, 3\}$ represent our designed prompt templates, and specific prompts are shown in the Appendix A.1. In practice, we use GLM-4[2] (Zeng et al., 2024) as $f$. Besides, "Core References" in Figure 1 means extracting papers referenced in introduction and method sections, because we believe these references have the greatest impact on paper $p$ among all references.

Additionally, the background, summary, and main ideas are also encoded with Sentence-BERT (Reimers & Gurevych, 2019) for their embeddings $e_b^{(p)}, e_s^{(p)}$ and $e_i^{(p)}$, respectively. All extracted information are recorded into our literature database.

---

[2] We use the GLM-4 released in May 20th, 2024 (glm4-20240520).

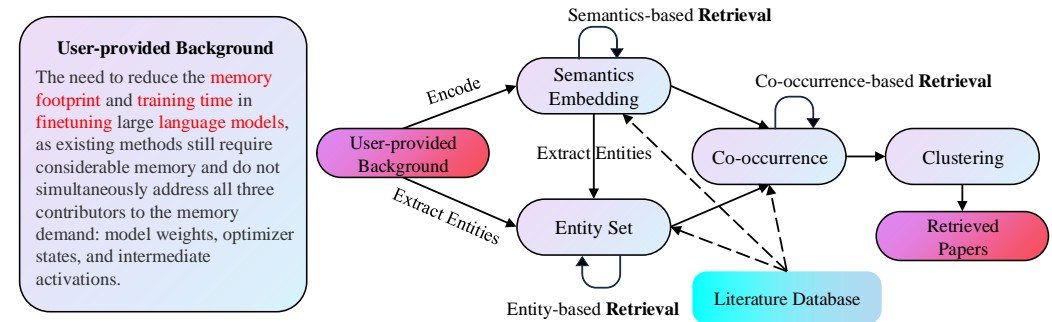

Figure 2: The pipeline of SEC-based literature retrieval and literature clustering. Red words in the user-provided background are entity examples.

To retrieve literature faster, we also construct a paper-entity graph in the database. we also store all occurrence relationships of papers and entities in the database. As shown in Figure 1, if an entity $T_{e1}$ appears in the paper $p1$, there will be an edge between the two paper nodes.

## 3.2 LITERATURE RETRIEVAL AND FILTERING

Literature retrieval is an essential process for idea proposal. It should follow the rule of comprehensiveness and low-redundancy. On the one hand, a comprehensive retrieval can provide researchers with instructive inspirations and avoid repetitive idea proposal. On the other hand, more retrieved papers are not necessarily better because redundant papers may also introduce noise and disperse a researcher's attention. To this end, we first propose a SEC-based (Semantics, Entities, and Citation co-occurrence) literature retrieval. Then, we propose a clustering-based literature filtering to pick out the most helpful papers. The process is shown in Figure 2.

### 3.2.1 SEC-BASED LITERATURE RETRIEVAL

**Semantics-based retrieval.** As shown in Figure 2, given a user-provided background $T_b^{(u)}$, we encode it as an embedding with Sentence-BERT (Reimers & Gurevych, 2019), marked as $e_b^{(u)}$. Then, $e_b^{(u)}$ is used to search in the literature database $\mathbb{D}$ for its semantic neighbors. Specifically, $e_b^{(u)}$ is compared with $e_b$ of all papers' backgrounds in the literature database to identify a subset of papers with the minimum cosine similarity as the semantic-based retrieval results. Assume the retrieved papers as $\mathbb{N}_1$,

$$\mathbb{N}_1 = \{p | e_b^{(p)} \in \text{TopK}(cosine(e_b^{(u)}, e_b^{(i)})) \text{ for } i \in \mathbb{D}\}, \tag{2}$$

where $p$ or $i$ represents a paper in the literature database. In practice, we take $K = 55$ for the TopK operation.

**Entity-based retrieval.** As we can see in Figure 2, after semantic literature retrieval, we take the user-provided background $T_b^{(u)}$ as input and prompt GLM-4 to extract all entities in the background. Then, the abstract section of semantics-based retrieved papers (*i.e.*, $p \in \mathbb{N}_1$) are also given to the GLM-4 to extract their entities. The exact prompt we use is provided in the Appendix A.1. After entity extraction, we also expand the entity set by giving these entities back to GLM-4 and let it generate some synonyms. The motivation behind entity expansion is that the same concept may express in different ways, and entity expansion can help us retrieve papers that use synonyms in the following process. We notate the entity set after synonym expansion as $\mathbb{E}_1$.

Additionally, we further expand the entity set through an entity-neighborhood-based approach. In simple terms, for an entity $T_e$ in the current entity set $\mathbb{E}_1$, any paper $p$ that includes entity $T_e$ should also have its other entities included in the candidate entity set. However, we find that this will induce many redundant or even noisy entities, and the reasons are twofold:

1. Two entities with low relevance may appear together in a paper due to the specific content requirements of that paper.

2. High-frequency words do not effectively characterize a paper or its background. For instance, the user-provided background might include the term "Transformer", but this does not imply that all entities co-occurring with "Transformer" in other papers are significant to us. This is because "Transformer" is a high-frequency term that may appear in many recent publications.

To this end, we propose two filtering mechanisms for neighborhood-based entity expansion:

1. An entity will only be supplemented if it has appeared together with another entity in at least $m$ papers. In practice, we take $m = 2$.

2. Inspired by the TF-IDF (Jones, 2004) algorithm, we believe that if an entity appears frequently across the entire paper database, it indicates that the entity is less representative. Therefore, we only select the $n$ entities that appear the least in all literature as the final entity set. In practice, we take $n = 5$.

The entity set after a second expansion is represented as $\mathbb{E}^{(u)}$. Entities are key words that are most relevant with a paper's topic. A paper is likely to be helpful to us if it contains entities that match those in our entity set $\mathbb{E}^{(u)}$. Thus, for any entity $T_e$ in set $\mathbb{E}^{(u)}$, we search for papers that also contain $T_e$ in our database. Marking all searched papers as a set $\mathbb{N}_2$,

$$\mathbb{N}_2 = \{p | \exists T_e \in \mathbb{E}^{(u)} \wedge T_e \in T_b^{(p)}, p \in \mathbb{D}\}. \tag{3}$$

**Co-occurrence-based retrieval.** In the above, we retrieve literature relevant to the user-provided background through entities and semantics. Wherein, entities represent specific details of a paper, while semantics represent the broader, overall meaning within the background. However, in actual research, we often encounter two papers, $p_1$ and $p_2$, which are neither similar in details nor in semantics, yet are cited together. This indicates that researchers have discovered a latent relationship between $p_1$ and $p_2$ in past studies. To capture and fully utilize these insights, we propose a literature retrieval method based on citation co-occurrence. Specifically, as shown in Figure 2, for any paper $p_1$ we have already retrieved, if $p_2$ is frequently cited alongside $p_1$ in other papers, we will include $p_2$ in our literature retrieval set:

$$\mathbb{N}_3 = \{p_2 | p_1 \in (\mathbb{N}_1 \cup \mathbb{N}_2) \wedge \text{co-cite}(p_1, p_2)\}, \tag{4}$$

where co-cite means $p_1$ and $p_2$ are often simultaneously cited by other papers. In practice, we select the 2 papers that are most frequently co-cited with each paper.

Finally, the whole retrieved papers can be represented as $\mathbb{N} = \mathbb{N}_1 \cup \mathbb{N}_2 \cup \mathbb{N}_3$.

### 3.2.2 LITERATURE CLUSTERING

After SEC-based literature retrieval, we may get over 500 papers, so further filtering is essential to pick out the most significant ones. Since we have observed that the retrieved papers often present similar ideas, we hope to retain only one paper among those with similar content during the generation of new ideas. To achieve this, we propose clustering the papers based on cosine similarity measures. Specifically, we first define the embedding of a retrieved paper as:

$$e^{(p)} = w_s e_s^{(p)} + w_i e_i^{(p)}, \tag{5}$$

where $e_s^{(p)}$ and $e_i^{(p)}$ are embeddings for summary and main ideas of an idea, as illustrated in Figure 1. We choose $w_s = w_i = 0.5$ in practice. Then, we apply clustering to group papers according to their cosine similarity. In practice, since the semantic embeddings of all papers are pre-recorded in a database, we only need to perform the similarity comparison and clustering processes. Finally, we select one paper from each cluster, respectively, and make up the retrieved papers.

### 3.3 IDEA PROPOSAL

Upon completion of the literature retrieval, we propose three approaches for generating research paper ideas. In essence, the idea generation process can leverage two types of information: the first is derived from the content of the retrieved papers, which inspires the LLM to generate ideas; the

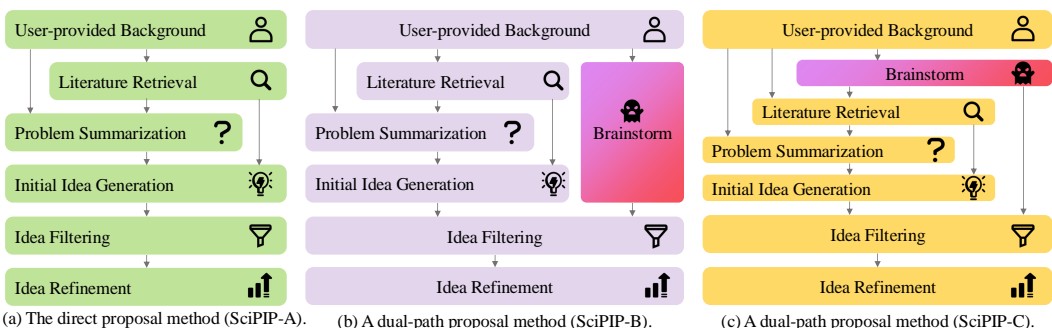

(a) The direct proposal method (SciPIP-A).     (b) A dual-path proposal method (SciPIP-B).     (c) A dual-path proposal method (SciPIP-C).

Figure 3: Three pipelines for idea proposal.

second involves the LLM freely brainstorming to produce new ideas. Based on this principle, we delineate three methods of idea generation that vary in their application of brainstorming.

As illustrated in Figure 3(a), the direct proposal method (SciPIP-A), does not use brainstorm. While the first dual-path proposal method (SciPIP-B), as Figure 3(b) shows, utilizes the user-provided background into two branches. The first branch employs this background for literature retrieval, problem summarization, and idea generation based on the retrieved literature, while the second branch engages in brainstorming solutions directly from the user-provided background. Following the independent generation of ideas in both branches, the outputs are merged and subsequently filtered and refined to yield the final ideas. Similarly, as shown in Figure 3(c), the second dual-path proposal method (SciPIP-C) follows a process analogous to SciPIP-B, with the key distinction being that the content generated through the LLM's brainstorming is utilized not only for idea generation but also integrated with the user-provided background for entity extraction and other literature retrieval processes. We will provide a detailed exposition of these three methods of idea proposal in the following sections. We use GPT-4o[3] by default in this section.

### 3.3.1 DIRECT IDEA PROPOSAL METHOD

As depicted in Figure 3(a), in the direct proposal method, we first retrieve papers following the pipeline described in Section 3.2. Then, the user-provided background along with the retrieved papers are utilized to prompt the LLM to summarize the core problem we aim to address and provide justifications. The specific prompts can be found in the Appendix A.1.

With the summarized problem and justifications, the LLM is prompted to generate around 10 initial ideas. In the prompt, both the problem, the justification and the retrieved papers are provided. The LLM is encouraged to generate clear, innovative, valid, and comprehensive ideas. The specific prompts for this step can be also found in the Appendix A.1.

Though the prompt has declared, the initially generated ideas may still have shortcomings in terms of novelty or relevance to the problem. To address this, we filter the initial ideas using prompt engineering (prompts are illustrated in the Appendix A.1), with the primary criterion being that the ideas are generated in response to the given problem. Additionally, the ideas must exhibit a high degree of novelty and feasibility. During this process, each generated idea is evaluated independently, and about half of them will be filtered.

Then, the LLM is encouraged to further improve the filtered ideas by considering their inter-relationships. That is, the LLM is tasked with considering the compatibility of the ideas, ensuring that it does not generate conflicting or repetitive ideas. Moreover, the LLM is required to generate formulas or algorithms to better elaborate the ideas if needed. The prompt is shown in the Appendix A.1. Finally, about 3 to 4 refined ideas will be proposed.

---

[3]We use the GPT-4o released in May 13th, 2024 (gpt-4o-2024-05-13), which has an October 2023 knowledge cutoff.

Table 1: The number of proposed ideas that successfully matched ACL 2024 ideas. More high-scoring ideas are better. "#" means "the number of". The results with $^\dagger$ are averaged over 1968 input backgrounds.

| Proposal Methods | Variants | #Backgrounds/ #Proposed Ideas | #Ideas of Similarity Score | | | | |
|---|---|---|---|---|---|---|---|
| | | | 4 | 3 | 2 | 1 | 0 |
| AI Scientist | - | 100 / 400 | 0 | 58 | 211 | 123 | 8 |
| SciPIP | SciPIP-A | 100 / 385 | 5 | 115 | 192 | 71 | 2 |
| | SciPIP-B | 100 / 379 | 4 | 139 | 157 | 75 | 4 |
| | SciPIP-C$^\dagger$ | 100 / 388 | 5 | 117 | 177 | 85 | 4 |
| | SciPIP-C | 1968 / 7638 | 91 | 2305 | 3492 | 1681 | 69 |

### 3.3.2 DUAL-PATH IDEA PROPOSAL METHODS

We find that the directly generated ideas often rely heavily on the retrieved literature, sometimes closely resembling the methods presented in those papers. They frequently involve transferring approaches from other fields or making minor improvements to existing methods within the same field, resulting in relatively ordinary novelty and rarely yielding breakthrough thinking.

Therefore, we further propose idea proposers that incorporates brainstorming, encouraging the LLM to produce more novel thoughts. Specifically, brainstorming can play a role in both processes of idea generation. As shown in Figure 3(b), the SciPIP-B has two paths, where one path follows the direct proposal approach, while the other path uses the LLM to brainstorm possible solutions based on the user-input background, outputting these as ideas. Ultimately, these ideas will be merged with those generated based on the retrieved papers, filtered and refined to produce the final ideas. In this model, the results of brainstorming are independent of the generation based on retrieved papers.

In another approach, as shown in Figure 3(c), brainstorming generates ideas independently while also being utilized in literature retrieval. Specifically, we extract entities from the brainstorming results and incorporate them as part of the entity set in the literature retrieval process. With this method, some keywords arising from the brainstorming will also help enhance the effectiveness of literature retrieval. The ideas generated through brainstorming will also be merged with those produced after literature retrieval.

## 4 EXPERIMENTS

### 4.1 EVALUATION DATASET

We collect all papers accepted by ACL 2024, including long papers, short papers, findings, and workshop papers. After excluding a few PDFs that could not be correctly parsed, 1,968 papers are remained for analysis. The remaining papers are processed similarly to those in the literature database in Section 3.1, with their entities, backgrounds, summaries, main ideas, detailed ideas, and references extracted in advance.

The experiments in this study are divided into two parts: retrospective experiments and innovation experiments. Retrospective experiments refer to testing whether different algorithms can generate the same ideas and literature retrieval results as the original papers on the evaluation dataset (i.e., ACL 2024 papers) with providing the background of the papers as input. In contrast, innovation experiments allow the models to freely propose new ideas, which are then evaluated from multiple perspectives, including novelty and feasibility.

### 4.2 RETROSPECTIVE EXPERIMENTS FOR IDEA PROPOSAL.

**Compared algorithms.** AI Scientist (Lu et al., 2024), when given an existing idea, iteratively refines the idea through multiple rounds of LLM inference. Afterward, the AI Scientist will expand the Idea into a full paper. Since our algorithm only focuses on proposing ideas, we only compare the idea proposal part with AI Scientist. For this purpose, we make slight adjustments to the AI Scientist's process. Specifically, for the user-provided background $T_b^{(u)}$, we first retrieve a paper from the literature database with a similar background. The idea from this paper serves as the initial

Table 2: The win rate of proposed ideas in terms of novelty and feasibility. The ideas are classified in terms of their similarity scores with their most similar existing ideas. The experiments are done on SciPIP-C proposed 7638 ideas.

| Similarity Score | 4 | 3 | 2 | 1 | 0 |
|---|---|---|---|---|---|
| Novelty | 10.2% | 13.1% | 16.4% | 20.1% | 40.2% |
| Feasibility | 19.1% | 11.5% | 16.7% | 25.5% | 23.2% |

Table 3: The novelty scores of proposed ideas. The scores are evaluated by GPT-4o after comparing with similar papers in Semantic Scholar.

| Proposal Methods | #Backgrounds/ #Proposed Ideas | #Ideas of Novelty Score | | | | | | | | | | |
|---|---|---|---|---|---|---|---|---|---|---|---|---|
| | | 10 | 9 | 8 | 7 | 6 | 5 | 4 | 3 | 2 | 1 | 0 |
| AI Scientist | 100 / 400 | 0 | 12 | 131 | 98 | 55 | 30 | 44 | 26 | 4 | 0 | 0 |
| SciPIP-A | 100 / 385 | 0 | 92 | 145 | 73 | 37 | 16 | 14 | 8 | 0 | 0 | 0 |
| SciPIP-B | 100 / 379 | 0 | 63 | 161 | 55 | 37 | 19 | 26 | 14 | 4 | 0 | 0 |
| SciPIP-C | 100 / 373 | 0 | 67 | 155 | 64 | 40 | 15 | 20 | 10 | 2 | 0 | 0 |

idea for refinement by the AI Scientist. In contrast, our algorithm directly uses the user-provided background $T_b^{(u)}$ as input for idea proposal. We then compare the similarity of generated ideas by two algorithms to the ideas from ACL 2024 papers.

**Evaluation Protocol.** To evaluate the matching rate between the generated ideas and those from ACL 2024, we first preprocess all ACL papers following the method in Section 3.1 and store them in a database. The generated ideas are then compared based on cosine distance to retrieve the 10 most similar ideas from the database. Next, using prompt engineering, GPT-4o selects the most similar idea and assigns a similarity score between 0 and 5, where a higher score indicates greater similarity. From our observations, a score of 4 indicates that the two ideas are almost identical, differing only in minor details, while a score of 3 or lower suggests more significant differences. Wherein, SciPIP-C is tested on all ACL 2024 papers, while other methods are tested with 100 backgrounds randomly sampled from the whole test set.

However, we believe that low-scoring ideas in the retrospective experiments do not necessarily lack value. On the contrary, some of these ideas exhibit strong novelty and feasibility, though they do not ideas published at ACL 2024. To further assess the novelty and feasibility of all ideas generated by SciPIP, we employ the LLMs for evaluation. For each round of comparison, we sample one idea from each of 5 similarity scores and ask the LLM to rank them based on their novelty and feasibility. We then record the win rate (*i.e.*, the probability of ranking first) of ideas across different similarity scores in all rounds.

**Results and analyses.** As we can see in Table 1, our proposed three idea proposal strategies can, on average, generate 4 to 5 ideas that highly match ACL 2024 conference papers out of every 100 input backgrounds. This indicates that SciPIP is capable of generating ideas consistent with human thought, whereas the highest similarity score for all ideas generated by the AI Scientist is only 3. Additionally, the three methods we propose exhibit similar performance.

Moreover, the results in Table 2 illustrates that ideas with lower similarities to published ideas even show higher novelty, while the reasons still need more explorations. Further, ideas do not show much difference in terms of their feasibility.

Besides, we also provide two examples of SciPIP proposed ideas in Figure 4. The two examples both get a similarity score of 4 to an existing paper in ACL 2024, and the generated idea is indeed very similar to the matched idea. For example, in the second example (with the yellow background), the background points out the drawback of existing code generation algorithms. Both our generated and the matched idea propose to iteratively refine the generated code, and reinforcement learning based reward model should be used to evaluate the generated code. The reward should be decided by the error resolution, the severity of errors, and so on. More examples can be seen in the Appendix A.2.

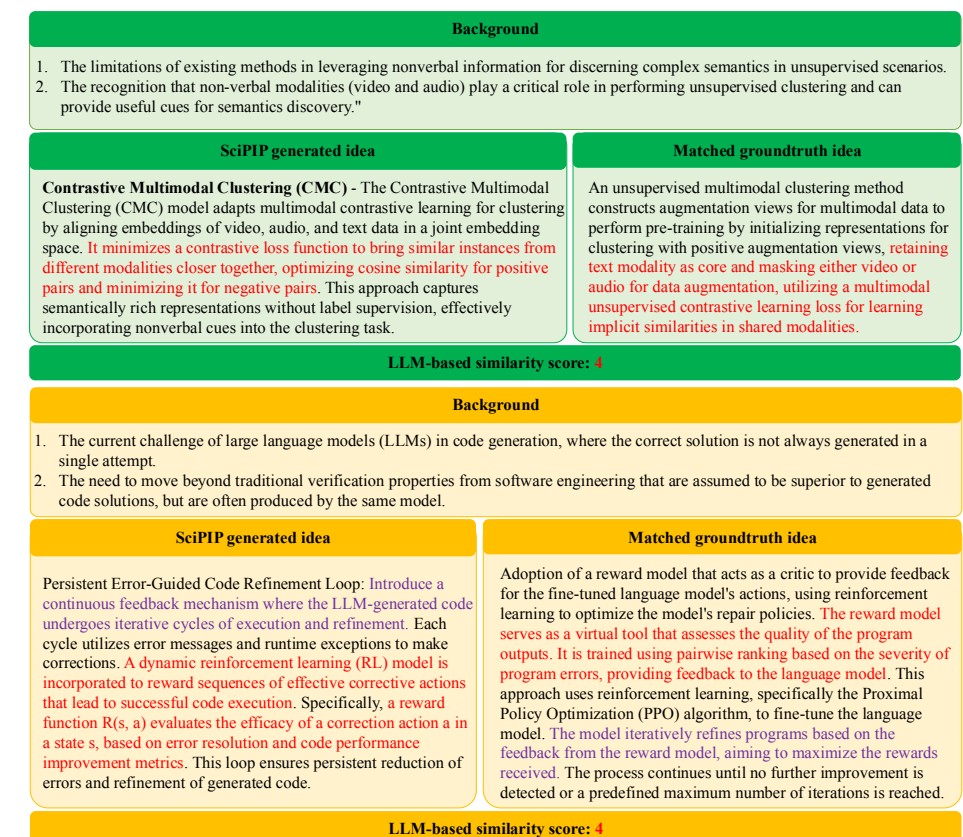

Figure 4: Randomly picked samples of SciPIP proposed ideas. Matched groundtruth idea means ideas proposed in some paper of ACL 2024.

### 4.3 NOVELTY EXPERIMENTS FOR IDEA PROPOSAL

**Compared algorithms and evaluation protocol.** We also compare with AI Scientist (Lu et al., 2024) for novelty verification. The verification way is drawn from the official source code of AI Scientist with some modifications. To be specific, a proposed idea will give some key words that being used to search similar papers in Semantic Scholar[4]. Through comparison with several similar papers drawn from Semantic ScholarGPT-4o judges the novelty of the generated idea. The novelty score is from 0 to 10, higher score means smaller similarity with existing papers or higher novelty.

**Results and analyses.** The results are in Table 3. It can be seen that both SciPIP and AI Scientist can generate very novel ideas with score 9. While our proposed ideas with 9 score are much more than AI Scientist (92 vs. 12). Unexpectedly, SciPIP with brainstorm perform worse than the direct proposal. It may be because brainstorm utilizes the knowledge from the GPT-4o itself in essence. Therefore, it is hard for the model to generate brand new ideas that are totally different with existing literature. However, we believe brainstorming will be a significant supplement to retrieval-based generation, so we still preserve the results of SciPIP-B/C, hoping attract the community's attention. At least, all versions of SciPIP generate over 270 high-scoring (score > 7) ideas even though they only match 4 to 5 ideas in ACL 2024. The results indicate that non-matching ideas may be more valuable because SciPIP generate novel ideas that do not appear (or even do not put forward by human).

### 4.4 RETROSPECTIVE EXPERIMENTS FOR PAPER RETRIEVAL

**Compared algorithms.** Since AI Scientist does not perform a literature retrieval when generating ideas, the results primarily on SCIMON (Wang et al., 2024) and ResearchAgent (Baek et al.,

---

[4]https://www.semanticscholar.org/

Table 4: The literature retrieval results. The groundtruth are the real citations of the tested papers. $Recall_{10}$ means the recall rate of the top 10 ranked papers among the retrieved literature compared to the ground truth citations.

| Retrieval Methods | $Recall_{10}$ | $Recall_{20}$ | $Recall_{30}$ | $Recall_{40}$ | $Recall_{50}$ |
|---|---|---|---|---|---|
| AI Scientist | | | Not Applicable | | |
| SCIMON-like | 0.381 | 0.481 | 0.548 | 0.587 | 0.616 |
| ResearchAgent-like | 0.377 | 0.484 | 0.550 | 0.598 | 0.622 |
| **SciPIP (Ours)** | **0.419** | **0.544** | **0.615** | **0.657** | **0.684** |

Table 5: Ablation studies for literature retrieval. SE means our proposed semantic-entity based retrieval, CC means citation co-occurrence, and CL means clustering.

| Semantics | Entity | SE | CC | CL | $Recall_{10}$ | $Recall_{20}$ | $Recall_{30}$ | $Recall_{40}$ | $Recall_{50}$ |
|---|---|---|---|---|---|---|---|---|---|
| ✓ | | | | | 0.377 | 0.484 | 0.550 | 0.598 | 0.622 |
| | ✓ | | | | 0.316 | 0.383 | 0.421 | 0.462 | 0.487 |
| | ✓ | | ✓ | | 0.348 | 0.428 | 0.468 | 0.506 | 0.529 |
| | | ✓ | | | 0.383 | 0.475 | 0.548 | 0.602 | 0.633 |
| | | ✓ | ✓ | | 0.391 | 0.497 | 0.576 | 0.624 | 0.668 |
| | | ✓ | | ✓ | 0.395 | 0.506 | 0.574 | 0.616 | 0.643 |
| | | ✓ | ✓ | ✓ | **0.419** | **0.544** | **0.615** | **0.657** | **0.684** |

2024). However, the experimental setups and literature database of SCIMON and ResearchAgent for generating scientific paper ideas differ from those in this study. Additionally, ResearchAgent is not open source, making it challenging to fully replicate the exact algorithm. Therefore, based on the descriptions in the original papers, we implement similar literature search algorithms, namely SCIMON-like and ResearchAgent-like in Table 4.

**Evaluation protocol.**   Only a few reference papers are crucial for generating a paper's idea; using all citations as ground truth may introduce significant noise. Among contemporaneous papers, there may be similar ideas, and researchers might only cite one of them. To address this, we propose two strategies: We believe that the most important citations for a paper typically appear in the introduction and method sections; thus, we extract only these sections' citations as ground truth during PDF parsing. Additionally, as mentioned earlier, our method clusters the retrieved literature after searching, treating all papers in the same cluster as similar. In the retrospective experiment, we evaluate the distance between ground truth citations and cluster centers. If a ground truth citation falls within a cluster retrieved by SciPIP, we consider the retrieval result correct.

**Results and analyses.**   The results are shown in Table 4, where $Recall_{10}$ represents the proportion of correctly retrieved papers when the algorithm is restricted to returning only 10 papers. For example, if the ground truth for a paper's literature search includes 20 references, a recall rate of 0.684 indicates that approximately 13 relevant papers were correctly retrieved. From the data in the table, it can be observed that our algorithm successfully retrieves more relevant papers compared to SCIMON and ResearchAgent. We also provide some ablation studies about literature retrieval in Table 5. As we can see, SE performs better than using only semantics or entities for retrieval. Moreover, citation co-occurrence and clustering also help improve the retrieval results.

# 5   CONCLUSIONS AND LIMITATIONS

In this paper, we propose a method for generating scientific paper ideas and demonstrate its effectiveness on natural language processing datasets. The experimental results show that SciPIP is capable of proposing numerous novel ideas through the capabilities of LLMs. These ideas not only match papers published at recent academic conferences but also exhibit significant potential in terms of novelty, feasibility, and other key aspects. Despite these positive results, we gain more questions than conclusions in this work. For example, why do the ideas with lower similarity score looks more novel (refereed as to Table 2). We need more explorations to answer these questions.

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

# A APPENDIX

## A.1 PROMPTS USED IN THIS PAPER

We employ prompt engineering accomplishing our task in this paper, and the used prompts are summarized in Table 6.

## A.2 EXAMPLES OF OUR GENERATED IDEAS

More examples of SciPIP proposed ideas are given in Figure 5.

Table 6: Summarization of our used prompts.

| Prompts | Place |
|---|---|
| The prompt for entity extraction, namely $\tau_1$. | Table 7 |
| The prompt for summary, background, and main ideas extraction, namely $\tau_2$. | Table 8 |
| The prompt for detailed ideas extraction, namely $\tau_3$. | Table 9 |
| The prompt for problem/rational generation. | Table 10 |
| The prompt for initial idea generation. | Table 11 |
| The prompt for idea filtering. | Table 12 |
| The prompt for idea improvement. | Table 13 |
| The prompt for brainstorming. | Table 14 |
| The prompt for picking out the most similar idea from several ideas. | Table 15 |
| The prompt for evaluating the similarity score between two ideas. | Table 16 |
| The prompt for scoring the novelty of an idea. | Table 17 |
| The prompt for comparing two ideas for their clarity, novelty, feasibility, and generalizability. | Table 18 |
| The prompt for comparing five ideas for their clarity, novelty, feasibility, and generalizability. | Table 19 |

Table 7: The prompt for entity extraction, namely $\tau_1$.

| | |
|---|---|
| **System Message** | Now you are an expert in extracting key entities from research contents. You are good at identifying the most important keywords or phrases that summarize the main topics or concepts discussed in the content. |
| **User Message** | Task Description:

I will provide you with a content from a research paper. Your task is to extract the key entities from this content. These entities are the most important keywords or phrases that summarize the main topics or concepts discussed in the content.

Instruction:

Content: The content is your key focus, and the extracted entities should be based on the content. In other words, the entities you extract should be concrete manifestations of the main themes and topics discussed in the content.

Your approach should be systematic:
- Start by thoroughly reading the content to understand its main themes and topics.
- Identify and list the key entities that are central to the content.
- Ensure that the entities are relevant, meaningful, and representative of the content.
- Each entity in entities should be no longer than 5 words.
- Each entity in entities should contain at least 2 words.
- The number of entities should be less than or equal to 5.
- Each entity in entities should be nouns or noun phrases.

examples:
{examples}

Your turn:
Given the following content:
{content}

Your answer should follow this format:
entity1, entity2, entity3, ...... |

Table 8: The prompt for summary, background, and main ideas extraction, namely $\tau_2$.

| | |
|---|---|
| **System Message** | Now you are an expert in extracting key entities from research contents. You are good at identifying the most important keywords or phrases that summarize the main topics or concepts discussed in the content. |
| **User Message For Summary** | Task Description:

You are provided with the title, abstract, and introduction of a research paper. Your task is to generate a concise summary of what kind of problem does this paper aim to solve and what methods are proposed to address it. The summary should follow this format: The problem of [problem] can be addressed by [main idea/approach].

Instructions:

Title: Read the title to understand the general topic of the paper. Abstract: Read the abstract to get a concise summary of the research, including the problem addressed, the methods used, and the main findings. Introduction: Read the introduction to gain a deeper understanding of the background, significance, and specific problem the paper addresses, as well as the proposed approach or solution. Based on the provided information, generate a single sentence that captures the essence of the paper, following the format specified above.

Your Turn:

Given the following paper information: Title: title Abstract: abstract Introduction: introduction

Output: The problem of [problem] can be addressed by [main idea/approach]. |
| **User Message For Background And Main Ideas** | Please read the title, abstract, and introduction of the paper again, as well as the summary you provided. Complete the following two tasks:
1.Briefly provide the two most critical motivations behind proposing these methods to address the problems.
2.Briefly provide the three most critical or innovative details of the paper that were not mentioned in your summary (It's best if these details are the new methods or techniques adopted in this paper).

Output:
Motivations:1.[motivation1]. 2.[motivation2]. Details:1.[detail1]. 2.[detail2]. 3.[detail3]. |

Table 9: The prompt for detailed ideas extraction, namely $\tau_3$.

| | |
|---|---|
| **System Message** | Now you are an expert in extracting key entities from research contents. You are good at identifying the most important keywords or phrases that summarize the main topics or concepts discussed in the content. |
| **User Message** | ### Task Description:
You will be provided with the abstract and a text extracted from a paper and three contributions of the paper. Your task is to filter, refine, and revise the content of the contributions through the text provided to you.

### Information Provided:
1. **Abstract**: It's the abstract directly extracted from the paper.
2. **Contributions**: These are the contributions (methods) we have summarized based on the abstract and introduction of the paper.
3. **Text**: It's the text directly extracted from the paper, containing the methodology of the paper.

### Approach:
Your approach should be systematic:
- **Step 1**: Start by reading the abstract and contributions, to understand the main work of this paper.
- **Step 2**: Then, read the text, to find information related to the contributions and ignore other information. If you think there is missing content in the contributions section, you can add one. On the contrary, if you think there is content duplication, merge or delete one. Please ensure that the final contributions have 2 to 4 entries.
- **Step 3**: Finally, provide specific details for each contribution as detailed and comprehensive as possible based on the content in the text. If applicable, you may include formulas or algorithms to support the ideas.

### Specific Information:
I will provide you with specific information now, please use them according to the instructions above:
1. **Abstract**: {abstract}
2. **Contribution**: {contribution}
3. **Text**: {text}

### Format for Your Response:
Your output should follow the format, and please note that your subject should not be 'the paper' but 'this method' or the specific method name:
**Idea 1**: [The first method idea]
- **Details**: [Details of the first idea]
**Idea 2**: [The second method idea]
- **Details**: [Details of the second idea]
... |

Table 10: The prompt for problem/rational generation.

| | |
|---|---|
| **System Message** | Now you are a researcher in the field of AI with innovative and pioneering abilities. You are good at proposing novel and valuable questions based on research background. |
| **User Message** | ### Task Description: You will receive a research background along with summaries, backgrounds, and contributions (methods) of several related papers. Your task is to carefully analyze this information and propose a research problem that is original, clear, feasible, relevant, and significant to its field. Additionally, provide the rationales behind the proposed problem. ### Information Provided: 1. **Research Background**: This is your primary focus. The research problem you propose should be a direct reflection of this background. 2. **Related Papers**: These papers offer studies directly related to the primary research topic, providing additional insights and knowledge that will inform your proposed problem. ### Approach: Your approach should be systematic: - **Step 1**: Begin by thoroughly understanding the core focus of the research background. - **Step 2**: Review the summaries, backgrounds, and contributions (methods) of the related papers to gain broader insights into the primary research topic. - **Step 3**: Based on the provided information, propose a research problem that meets the criteria of being original, clear, feasible, relevant, and significant. Support your problem statement with clear rationales. ### Specific information: I will provide you with specific information now, please use them according to the instructions above: 1. **Research Background**: {background} 2. **Related Papers**: {related_papers_information} ### Format for Your Response: **Research Problem**: [your problem] - **Rationales**: [the rationale behind your problem] |

Table 11: The prompt for initial idea generation.

| System Message | Now you are a researcher in the field of AI with innovative and pioneering abilities. You are good at using innovative and original methods to solve cutting-edge problems in the field of AI. |
| --- | --- |
| User Message | ### Task Description:
You will be provided with a research problem along with its rationales. Your task is to brainstorm some ideas that are clear, innovative, valid, and comprehensive to address the problem. Additionally, some cue words along with summaries, backgrounds, and contributions (methods) of related papers will be provided as sources of inspiration for generating novel ideas.
### Information Provided:
1. **Research Problem & Rationales**: The key issues or aspects of the problem that need to be addressed. These will form the foundation for generating your ideas.
2. **Related Papers**: Draw inspiration from the abstracts, backgrounds, and methods of these papers. Delve deeply into these methods, understand the motivations behind them, and think critically about how they might inform your approach. Avoid merely stacking existing methods; instead, integrate relevant aspects with your own insights to create original solutions.
### Approach:
Your approach should be systematic:
- **Step 1**: Thoroughly read the research problem to understand your primary focus.
- **Step 2**: Review the summaries, backgrounds, and contributions (methods) of the related papers to gain a broader perspective and insights relevant to the problem.
- **Step 3**: Based on the provided information, propose some ideas that are clear, innovative, valid, and comprehensive.
### Specific Information:
I will provide you with specific information now, please use them according to the instructions above:
1. **Research Problem & Rationales**: {problem}
2. **Related Papers**: {related$_p$apers$_i$nformation}
### Format for Your Response:
Please ensure that your final ideas include about 10 entries, presented in the following format:
**Idea 1**: [The first method idea]
**Idea 2**: [The second method idea]
**Idea 3**: [The third method idea]
... |

Table 12: The prompt for idea filtering.

| | |
|---|---|
| **System Message** | Now you are a researcher in the field of AI. You are good at selecting the ideas that meet the requirements. |
| **User Message** | ### Task Description:
You will be provided with some ideas you previously generated, and a research background. Your task is to select 5-6 ideas that best address the problems described in the research background (priority) and ideas that are relatively novel and feasible (secondary).
### Information Provided:
1. **Ideas**: These are the ideas you previously generated based on the research background and several related papers.
2. **Research Background**: This document describes specific problems and challenges that need to be addressed.
### Approach:
Your approach should be systematic:
- **Step 1**: Analyze the research background to understand the specific problems that need solutions.
- **Step 2**: Critically review the ideas, selecting 5-6 ideas that are most effective in solving the problems in the research background (priority) and that are also relatively novel and feasible (secondary).
### Specific Information:
I will provide you with specific information now; please use them according to the instructions above:
1. **Ideas**: {idea}
2. **Research Background**: {background}
### Format for Your Response:
Please ensure that your final ideas include 5-6 entries, whose content has not been modified. Don't generate any explanation and just present the filtered ideas as well as their content in the following format:
**Idea 1**: [The first method idea]
**Idea 2**: [The second method idea]
**Idea 3**: [The third method idea]
... |

Table 13: The prompt for idea improvement.

| | |
|---|---|
| **System Message** | Now you are a researcher in the field of AI with innovative and pioneering abilities. You are good at using innovative and original methods to solve cutting-edge problems in the field of AI. |
| **User Message** | ### Task Description:
You will be provided with the research background and the original ideas you previously generated. Your task is to refine these original ideas by filtering out those with low feasibility and insufficient novelty while enhancing the most critical and relevant ideas to make them more novel, feasible, targeted, and specific. If applicable, you may include formulas or algorithms to support the ideas. Additionally, please adhere to the following requirements:
1. Do not generate ideas that are repetitive or contradictory.
2. Ensure that the generated ideas are coherent and form a cohesive whole.
### Information Provided:
1. **Research background**: This is the starting point of the original idea and the basis for analyzing whether the idea should be filtered.
2. **Original ideas**: These are the ideas you previously generated based on research background and several related papers.
### Approach:
Your approach should be systematic:
- **Step 1**: Thoroughly review the research background to understand the context and objectives.
- **Step 2**: Analyze the original ideas critically, identifying aspects with low feasibility or insufficient novelty, and then filter out them.
- **Step 3**: Enhance the most critical and relevant ideas by making them more novel, feasible, targeted, and specific. Incorporate formulas or algorithms if they strengthen the ideas.
### Specific Information:
I will provide you with specific information now, please use them according to the instructions above:
1. **Research background**: {background}
2. **Original idea**: {idea}
### Format for Your Response:
Please ensure that your response only includes the final ideas, which include 2 to 4 entries, presented in the following format:
**Idea 1**: [The first method idea]
- **Details**: [Details of the first idea]
**Idea 2**: [The second method idea]
- **Details**: [Details of the second idea]
... |

Table 14: The prompt for brainstorming.

| **System Message** | Now you are a researcher in the field of AI with innovative and pioneering abilities. You are good at generating creative and original ideas. |
|---|---|
| **User Message** | ### Task Description:
You are an AI researcher tasked with brainstorming initial, innovative ideas to address a given research problem in AI. Focus on generating diverse and creative approaches rather than finalized methods. The ideas can be rough and in their infancy but should cover a range of possible directions that could be explored further.

### Information Provided:
- **Research Background**: {background}

### Approach:
Your brainstorming should be systematic:
- **Step 1**: Thoroughly understand the research background.
- **Step 2**: Generate a list of 4 to 6 high-level ideas or directions that could potentially solve problems in the given background. Be creative, think outside the box, and avoid merely rephrasing existing methods.

### Format for Your Response:
Please present 4 to 6 ideas in the following format:
**Idea 1**: [Brief description of the first idea]
**Idea 2**: [Brief description of the second idea]
... |

Table 15: The prompt for picking out the most similar idea from several ideas.

| **System Message** | - |
|---|---|
| **User Message** | ### Task Description:
You will be provided with an idea you previously generated, and some reference ideas. Your task is to select the idea that is most similar to the one you generated from the reference ideas.

### Information Provided:
1. **Generated Idea**: This is the idea you previously generated based on research background and several related papers.
2. **Reference Ideas**: These are the ideas that you should select from.

### Approach:
Your approach should be systematic:
- **Step 1**: Analyze the generated idea to understand the methods it describes.
- **Step 2**: Critically review the reference ideas, selecting the idea that is most similar to the methods in the generated idea.

### Specific Information:
I will provide you with specific information now, please use them according to the instructions above:
1. **Idea**: {idea}
2. **Reference Ideas**: {reference_ideas}

### Format for Your Response:
Your answer can only have one number (strating from 1), indicating the number of the most similar idea, and cannot contain any other content. |

Table 16: The prompt for evaluating the similarity score between two ideas.

| **System Message** | - |
|---|---|
| **User Message** | ### Task Description: 
 You will be provided with an idea you previously generated, and a reference idea. Your task is to determine the similarity between the generated idea and the reference idea and give a score from 0 to 5. 

 ### Information Provided: 
 1. **Generated Idea**: This is the idea you previously generated based on research background and several related papers. 
 2. **Reference Idea**: This is the idea we provide you with that you need to compare with the generated idea. 

 ### Approach: 
 You should follow the following scoring criteria: 
 - **0**: The generated idea and reference idea are completely unrelated with no discernible similarities. 
 - **1**: The generated idea and reference idea have a vague connection, but differ significantly in their main concepts or approach. 
 - **2**: The generated idea and reference idea share a general concept but differ in most key aspects such as methodology or application. 
 - **3**: The generated idea and reference idea are similar in several areas, including general concept and some aspects of methodology, but differ in details or specific approaches. 
 - **4**: The generated idea and reference idea are largely similar in concept, methodology, and approach, with only minor differences in specifics. 
 - **5**: The generated idea and reference idea are nearly identical in all key aspects, including concept, methodology, and approach. 

 ### Specific Information: 
 I will provide you with specific information now, please use them according to the instructions above: 
 1. **Generated Idea**: {idea} 
 2. **Reference Idea**: {reference_idea} 

 ### Format for Your Response: 
 Your answer can only have one number (from 0 to 5), indicating the similarity score, and cannot contain any other content. |

Table 17: The prompt for scoring the novelty of an idea.

| | |
|---|---|
| **System Message** | You are an ambitious AI PhD student who is looking to publish a paper that will contribute significantly to the field.
You have an idea and you want to check if it is novel or not. I.e., not overlapping significantly with existing literature or already well explored.
Be a harsh critic for novelty, ensure there is a sufficient contribution in the idea for a new conference or workshop paper.
You will be given access to the Semantic Scholar API, which you may use to survey the literature and find relevant papers to help you make your decision.
The top 10 results for any search query will be presented to you with the abstracts.

You will be given num$_r$oundsroundstodecideonthepaper.
At any round, compare the provided idea with the information found in the article and provide a novelty score from 0 to 10.
In each search round, you should give a query and a novelty score based on the information in the relevant papers.
If there are no relevant papers, give a novelty score based on your own feelings. |
| **User Message** | Round current_round/num_rounds.
You have this idea:

"idea"

The results of the last query are (empty on first round):
"last_query_results"

Respond in the following format:

THOUGHT:
<THOUGHT>

RESPONSE:
''' json
<JSON>
''''

In <THOUGHT>, first briefly reason over the idea and identify any query that could help you suggest a score based on its novelty. Then give your perceived novelty score.

In <JSON>, respond in JSON format with ONLY the following field:
- "Query": An optional search query to search the literature (e.g. attention is all you need). You must make a query if you have not decided this round.
- "Novelty Score": A novelty score from 0 to 10.

A query will work best if you are able to recall the exact name of the paper you are looking for, or the authors.
This JSON will be automatically parsed, so ensure the format is precise.
(the JSON MUST contain the "Query" and the "Novelty Score")
In the last round, you should assign a "" value to the "Query" even if you don't need to generate it. |

Table 18: The prompt for comparing two ideas for their clarity, novelty, feasibility, and generalizability.

| | |
|---|---|
| **System Message** | You are an artificial intelligence researcher with extensive knowledge in this field, and now you need to make a comprehensive comparison between two ideas.
You will obtain a comparison standard, compare every point on the standard, and make a summary comparison at the end. |
| **User Message** | ### Comparison Standard:
" " "
**Clarity**: It evaluates whether the method is articulated in a straightforward and coherent manner, facilitating a comprehensive understanding for both practitioners and researchers, thus enabling effective application and potential adaptation in similar studies.
**Novelty**: It assesses the degree to which the method presents novel ideas or transformative strategies that challenge conventional practices, fostering advancements in the field and inspiring future research directions.
**Feasibility**: It examines the practicality and implementability of the method, ensuring that the required resources, time, and expertise are realistically available for its execution within the constraints of the study environment.
**Generalizability**: It determines how broadly the method can be extended or adapted to various contexts, populations, or situations, evaluating its applicability beyond the specific conditions of the study while maintaining relevance and effectiveness.
" " "

### You should compare these two ideas:
" " "IDEA1
idea1
" " "

" " "IDEA2
idea2
" " "

### Respond in the following format:

THOUGHT:
<THOUGHT>

RESPONSE:
'''json
<JSON>
'''

In <THOUGHT>, You can record your reasoning process to make your comparison more organized..

In <JSON>, respond in JSON format with ONLY the following field:
- "Clarity": Choose between 1 and 2 (If idea1 is better, fill in 1; otherwise, fill in 2. The same applies below.)
- "Novelty": Choose between 1 and 2
- "Feasibility": Choose between 1 and 2
- "Generalizability": Choose between 1 and 2
- "summary": Choose between 1 and 2
This JSON will be automatically parsed, so ensure the format is precise. |

Table 19: The prompt for comparing five ideas for their clarity, novelty, feasibility, and generalizability.

| | |
|---|---|
| **System Message** | You are an artificial intelligence researcher with extensive knowledge in this field, and now you need to make a comprehensive comparison among five ideas.
You will obtain a comparison standard, compare every point on the standard, and make a overall ranking at the end. |
| **User Message** | ### Comparison Standard:
" " "
**Clarity**: It evaluates whether the method is articulated in a straightforward and coherent manner, facilitating a comprehensive understanding for both practitioners and researchers, thus enabling effective application and potential adaptation in similar studies.
**Novelty**: It assesses the degree to which the method presents novel ideas or transformative strategies that challenge conventional practices, fostering advancements in the field and inspiring future research directions.
**Feasibility**: It examines the practicality and implementability of the method, ensuring that the required resources, time, and expertise are realistically available for its execution within the constraints of the study environment.
**Generalizability**: It determines how broadly the method can be extended or adapted to various contexts, populations, or situations, evaluating its applicability beyond the specific conditions of the study while maintaining relevance and effectiveness.
" " "

### You should compare these five ideas:
" " "IDEA1
idea1
" " "
" " "IDEA2
idea2
" " " ...
### Respond in the following format:

THOUGHT:
<THOUGHT>

RESPONSE:
'''json
<JSON>
'''

In <THOUGHT>, You can record your reasoning process to make your comparison more organized..

In <JSON>, respond in JSON format with ONLY the following field:
- "Clarity": Provide an array consisting of 1-5, representing each idea separately, with the better idea placed at the beginning (e.g. [4, 5, 3, 2, 1])
- "Novelty": Same as above.
- "Feasibility": Same as above.
- "Generalizability": Same as above.
- "Overall Ranking": Same as above.

This JSON will be automatically parsed, so ensure the format is precise. |

**Background**

1. The need to reduce hallucinated content in responses from LLM-based chatbots, which limits their reliability in sensitive domains like healthcare and education.
2. The requirement for a training-free and easy-to-use method that can improve the credibility of chatbot responses without additional data annotation or extensive retraining.

| SciPIP generated idea | Matched groundtruth idea |
|---|---|
| **Idea 1**: LIME-Based Post-Processing Filter with Reinforced Logical Verification **Details:** Enhance the basic LIME-based post-processing filter by integrating a reinforcement mechanism that goes beyond mere flagging of suspect segments. After LIME evaluates which parts of the response are grounded in factual data and which may be hallucinated, an additional logical verification step is applied using a lightweight, training-free logical consistency checker. This checker could employ principles from SATNet to ensure the logical coherence of the flagged segments. If logical inconsistencies are detected, the response can either be discarded, revised, or accompanied by a disclaimer highlighting the possibly unreliable segments. | Step-by-step verification protocol for reasoning chains. This protocol formalizes the process of verifying the correctness of each step in a reasoning chain, including the relevance of each step, the attribution of each step to external sources, and the logical correctness of each step, enabling fine-grained evaluation of reasoning verifiers.. |

**LLM-based similarity score: 4**

**Background**

1. The need to improve the effectiveness of large language models in utilizing retrieved information for enhanced text generation in retrieval-augmented generation frameworks.
2. The absence of a clear training mechanism that teaches LLMs to refine and integrate knowledge from retrieved texts of varying quality.

| SciPIP generated idea | Matched groundtruth idea |
|---|---|
| **Contrastive Learning for Enhanced Knowledge Integration**: Employ contrastive learning techniques to train the LLM using pairs of relevant and irrelevant retrieval instances, with a contrastive loss function $L_{\text{contrastive}} = d_{\text{pos}} - d_{\text{neg}}$. This approach helps the model distinguish between high-quality and low-quality information, improving the integration of the most informative and pertinent knowledge for text generation. | A method for generating training data using Large Language Models (LLMs) that incorporates both positive and negative examples and employs a contrastive loss objective. This method leverages LLMs to generate a diverse set of training examples, which includes both positive utterances related to an intent and negative utterances that are unrelated or express a negation of the intent. By using a contrastive loss term, the model is encouraged to encode similar semantic meanings closer together in the embedding space, while pushing apart embeddings of differing or opposite meanings. This approach aims to enhance the model's semantic encoding capabilities, particularly for understanding negations and implicatures. |

**LLM-based similarity score: 4**

**Background**

1. The need to bridge the gap between the capabilities of current benchmarks and the real-world visually grounded tasks that require agents to process both visual and textual information.
2. The desire to advance the development of autonomous agents by providing a rigorous assessment that simulates human interaction with modern computing interfaces.

| SciPIP generated idea | Matched groundtruth idea |
|---|---|
| **Interactive Contextual Scenario Emulation** **- Details**: Design advanced contextual scenarios that require agents to interact dynamically with both text and visual content. These scenarios should emulate real-world tasks such as web browsing, multimedia interpretation, and virtual assistants navigating through user instructions. Implement a simulation environment where these tasks evolve based on the agent's interactions, measuring adaptability and decision-making in context-rich environments. The evaluation metrics should include task completion accuracy, adaptability to new information, and responsiveness to real-time changes, simulating human-computer interaction dynamics more rigorously. | Contribution 1: Introduction of VisualWebArena, a Multimodal Benchmark for Web-based Tasks Details: This method introduces a novel benchmark suite called VisualWebArena, which is specifically designed to evaluate the performance of autonomous multimodal agents on visually grounded web tasks. It comprises a diverse set of web environments including Classifieds, Shopping, and Reddit. These environments contain realistic tasks that demand agents to process and understand image-text inputs, interpret natural language instructions, and execute actions on websites to achieve predefined objectives. |

**LLM-based similarity score: 4**

Figure 5: Some more randomly picked samples of SciPIP proposed ideas. Matched groundtruth idea means ideas proposed in some paper of ACL 2024.

