# OpenReview forum: "SciPIP: An LLM-based Scientific Paper Idea Proposer"
_ICLR.cc/2025/Conference — ICLR 2025 Conference Withdrawn Submission_

### Official Review · Reviewer_JoFd · 2024-10-25

**Soundness:** 2
**Presentation:** 3
**Contribution:** 3
**Rating:** 5
**Confidence:** 2

**Summary:**

This paper introduces a system called SciPIP, designed to help researchers come up with new ideas more effectively.
SciPIP starts by retrieving relevant studies from a research database, using that information to generate inspiring concepts.
It then goes a step further by leveraging a model to brainstorm entirely new ideas.
This approach helps SciPIP strike a balance between novelty and feasibility. Experimental results show that SciPIP not only identifies ideas similar to existing research but also generates promising, innovative concepts.

**Strengths:**

The system uses Semantic, Entity, and Citation co-occurrence retrieval techniques to ensure that the retrieved literature covers multiple dimensions of information.

SciPIP combines literature retrieval and brainstorming to generate ideas.

The system filters and refines the initially generated ideas.

**Weaknesses:**

Overall, this is an interesting study. However, I still have some concerns:

Large language models may rely on knowledge from their training data when generating ideas. If the model has "seen" content similar to existing research, it could end up producing ideas that are essentially variations of what’s already out there. So, a high similarity score might just mean the model is "copying" known information, rather than truly generating something new.

In many cases, the model’s similarity scores don’t give enough insight to tell if an idea is truly innovative. A high similarity score might just mean the idea makes minor tweaks to existing work, while a low score could indicate it’s different—but if it doesn’t align with the actual research question, that difference isn’t useful. This makes the use of similarity scores quite limited, and it calls for other tools to fill in the gaps.

Since similarity scores are based on semantic overlap, they don’t really measure the quality of the idea itself. Semantic similarity only captures how similar the wording is, not the underlying logic, originality, or technical feasibility. Relying on these scores alone could lead to conclusions that miss the actual value of the ideas.


One potential concern is the reliability of using large language models (LLMs) to evaluate the originality of ideas generated by SciPIP. Since LLMs may have been trained on a vast corpus that includes the same or similar literature, there could be a risk of generating ideas that are not genuinely novel. This raises questions about the ability of LLMs to accurately assess the originality and publishability of these generated ideas. Given this limitation, would it be more effective to supplement the evaluation with alternative methods, such as expert reviews? These approaches could help to ensure that SciPIP’s outputs demonstrate true innovation and relevance to the research community.

The current evaluation mainly relies on comparing generated ideas to existing papers using similarity scores, but this approach has some limitations in assessing true novelty and academic value.
A high similarity score doesn’t necessarily mean the idea lacks innovation, and a low score doesn’t always indicate a breakthrough.
Since the similarity score is based on semantic matching, it may miss aspects like technical feasibility or practical relevance, making it hard to capture the full academic contribution.
Additionally, without user studies or human expert evaluation, it’s challenging to verify the system’s real-world applicability, which may weaken its persuasive power.
This could make publishing as a research paper more challenging, especially for venues that prioritize metrics of academic innovation. Positioning it as a demo paper or incorporating more user studies could better highlight the system’s practical usefulness and potential impact.


it would be better to cite and discuss the following paper:
Li, Weitao, et al. "Citation-Enhanced Generation for LLM-based Chatbot." arXiv preprint arXiv:2402.16063 (2024).

Would this paper be more suitable as a demo paper?
Alternatively, would it be better to conduct additional user studies and shift toward an HCI-focused direction?
Given the current evaluation approach, it seems that positioning it as a research paper may be more challenging.

**Questions:**

Could a high similarity score simply mean the model is “copying” known information rather than truly innovating?
And might the reliance on semantic similarity scores lead to misjudging the actual value of an idea?

Measuring innovation is tough, so I still have some concerns. Is the range of applications for similarity scores limited?
In many cases, a high score might just reflect minor tweaks to existing work, while a low score could show a difference from current literature—but if it’s not relevant to the research question, that difference might not be valuable either.

---

### Official Review · Reviewer_6bY3 · 2024-10-31

**Soundness:** 2
**Presentation:** 3
**Contribution:** 2
**Rating:** 3
**Confidence:** 4

**Summary:**

This paper introduces SciPIP, an automated system for literature retrieval and scientific idea generation. In the literature retrieval phase, SciPIP considers semantic, topical, and citation-based factors to identify relevant papers and employs clustering to further filter and select the most significant literature. In the subsequent idea generation phase, SciPIP uses a dual-path proposal method, prompting the LLM to generate ideas both with and without augmented knowledge, then combines the results to enhance idea diversity and depth.

Experimental results show that SciPIP outperforms both AI Scientist in idea proposal capability and SciMON in literature retrieval effectiveness. Both AI Scientist and SciMON are existing frameworks that leverage LLMs for scientific idea generation, highlighting SciPIP’s advancements in this area.

**Strengths:**

1. The topic of the paper is interesting and important. The idea of using LLM to generate new scientific ideas is a critical research direction in understanding the ability of LLM and is interesting to have further exploration.

2. The proposed framework demonstrates notable improvements over a previous baseline and introduces several techniques that enhance both the literature retrieval and idea generation modules. In the literature retrieval module, SciPIP creates three distinct embeddings for the Background, Summary, and Main Ideas of each paper, rather than using a single embedding per paper. These embeddings are utilized differently across retrieval stages to improve accuracy. In the idea generation module, SciPIP goes beyond simply augmenting the LLM with retrieved literature by prompting the LLM to generate ideas directly. These two outputs are then aggregated and refined to enhance the final generated content.

3. The paper’s presentation is clear and accessible, with visual aids like Figures 1 and 3 effectively illustrating the proposed methods.

**Weaknesses:**

1. In SciMON, a related paper on scientific idea discovery, semantic, entity, and citation signals are also leveraged during literature retrieval. Although SciPIP extends SciMON's framework with additional techniques and demonstrates improved performance, the enhancements are generally minor and come with extra computational costs and complexity.

2. While the three pipelines for idea generation are interesting, they lack a clear explanation of the underlying intuition for each submodule. An ablation study is also missing, leaving unclear how each submodule contributes to the overall performance.

3. The evaluation pipeline raises further concerns. For instance, in assessing similarity with ACL papers, it’s unclear how well an LLM can judge similarity between ideas, given that ideas can align across different dimensions—such as background, settings, or methods. Simply prompting the LLM to provide a similarity score appears insufficient. Additionally, the paper outlines several modules necessary to prompt LLMs to generate reliable research ideas. Thus, it seems unlikely that LLMs alone can accurately determine novelty, suggesting that scoring generated ideas solely based on a novelty score from an LLM may not be reliable.

4. The reviewer also questions the quality of generated ideas. For example, in Figure 4, SciPIP generates an idea for multimodal learning that involves “minimizing a contrastive loss function to bring together similar instances of different modalities,” a widely accepted approach in the field (e.g., CLIP [1], published in 2021). Furthermore, the matched ground-truth idea discusses data augmentation, a concept omitted in the generated idea, yet the LLM assigns a similarity score of 4.

[1] Radford, A., Kim, J. W., Hallacy, C., Ramesh, A., Goh, G., Agarwal, S., ... & Sutskever, I. (2021, July). Learning transferable visual models from natural language supervision. In International conference on machine learning (pp. 8748-8763). PMLR.

**Questions:**

Among all the weaknesses mentioned above, the reviewer is mainly concerned about whether relying on an LLM as an evaluator for scientific ideas is sufficiently robust, and several questions must be considered:

1. How accurately do LLMs assess the similarity and novelty of a paper compared to human evaluators?

2. To what extent do their assessments differ from human judgments?

3. Is there consistency in LLM scoring across various contexts and paper types?

4. Could the LLM evaluator exhibit bias against ideas derived from related work, especially given that SciPIP leverages related work while AI Scientist does not?

---

### Official Review · Reviewer_Qkgm · 2024-11-02

**Soundness:** 3
**Presentation:** 2
**Contribution:** 1
**Rating:** 3
**Confidence:** 4

**Summary:**

The paper introduces the Scientific Paper Idea Proposer (SciPIP), a tool designed to assist researchers in overcoming the challenges posed by information overload and enhancing the novelty of interdisciplinary research. This system employs a hybrid approach: leveraging large language models (LLMs) to suggest novel research ideas based on retrieved literature, which is sourced using a new retrieval method. The proposed dual-path strategy aims to balance feasibility with originality in idea generation.

**Strengths:**

The paper is in line with the current trend of looking for the automation of research fostered by AI. The original part of the work is represented mainly by the multi-path proposal.

**Weaknesses:**

Many aspects of the evaluation are delegated to GPT-4o. It is not clear how, for instance, "novelty" and "feasibility" are determined except that GPT-4o is used for the evaluation.

Also, SentenceBERT may not be the best option to evaluate the similarity between entities and textual fragments, especially if the pre-computed models are used. It is known that SentenceBERT has some limitations when comparing nuanced semantics.

The paper's approach to aid researchers through enhanced literature retrieval and idea suggestion is noteworthy but not path-breaking. It represents a solid application of existing techniques like LLMs and embedding-based retrieval methods but lacks a transformative innovation to elevate its contribution to the research community.

Overall, while useful, SciPIP as presented currently may be best suited for practical use in specific contexts rather than serving as a major research contribution.

Some parts of the paper are not detailed well enough and make it difficult to follow the understanding of the paper (see questions).

**Questions:**

- 243:244 this could be addressed by recurring to Latent Semantic Indexing maybe?
It seems you are evoking the LSI principle but without using it explicitly.

- Table 1 is not commented and frankly I don’t understand what it is showing, what are the different scores and what they indicate. Even after reading the following paragraph I understand only that it is GPT-4o outputting these scores.

---

### Official Review · Reviewer_3Kwp · 2024-11-04

**Soundness:** 2
**Presentation:** 2
**Contribution:** 1
**Rating:** 3
**Confidence:** 4

**Summary:**

This paper explores an emerging and interesting application of LLM in proposing scientific ideas. The paper creates a new annotated database of 50k recent AI/ML papers (annotated with entities and embeddings for ideas, backgrounds, summary etc). The paper proposes a multi-pronged literature retrieval strategy using entity retrieval, semantic retrieval, co-citation retrieval and clustering. This specific ensemble strategy is new, to the best of my knowledge. The paper then proposes three pipelines for idea proposal that includes idea generation, filtering, refinement and independent brainstorming using gpt-4o. The experiment setting is using an NLP conference's recent publications. In the experiment, the paper compares its own retrieval strategy with two other retrieval strategies, and observed improved performance. The paper also compares its idea proposal strategy with another strategy (i.e. AI Scientist), based on gpt-4o's judgments, the paper's method outperforms its counterpart in terms of novelty, feasibility etc.

**Strengths:**

1. The paper is generally well-written and easy to follow. The method and experiment sections are clear.

2. The paper creates a new dataset with 50k AI/ML papers. I think it is a nice touch to annotate each paper and include entities such as main ideas, backgrounds. To my knowledge, no comprehensive scientific literature database exists where each paper's ideas are extracted and annotated.

3. The paper builds an LLM-powered system for idea proposal, including idea generation, filtering, refinements, and coupled with an independent LLM brainstorming module. I am fairly certain this system outperforms vanilla approach to ask LLM to generate new research ideas, at least in the field of NLP.

**Weaknesses:**

1. I am very concerned about the insufficient related work section. It is very short and does not cover many important aspects. For example, there are many other existing scientific literature database, arXiv, pmc, pubmed, semantic scholar, with much larger scale (over millions of articles), but they are not cited. There have been many advancements in the field of neural information retrieval with state-of-the-art embedding models such as OpenAI-v3, Grit-LM, Rank-Llama, but the authors choose to use a fairly out-of-date embedding model (sentence-bert) with a much shorter context window. There are also many works on using LLM directly to retrieve, search and select scientific literature such as RankGPT, this method is compared with other GPT-based retrieval strategies in BIRCO, a benchmark that includes a scientific literature benchmark, DORIS-MAE. None of these works are mentioned. At least some of them should be used as baseline models to compare with the paper's proposed retrieval strategy.

2. The second concern is the scale and comprehensiveness of this work. This work only focuses on a very narrow field in computer science, namely AI/ML, and its experiment setting is on an even narrower field, NLP. I would be fine with these settings except for the fact that the authors named their paper "scientific paper idea proposer". The word scientific implies a much larger domain, including biomedicine, physics, math, computer science and other quantitative sciences. Because the scale of this work is severely limited, I am doubtful for its contribution to the research community. FYI, these limitations about scales and lack of comprehensiveness should be openly acknowledged in the limitation section (which is also insufficient and very short).

3. The third concern is the lack of originality of this work. It is true that the problem this paper is trying to tackle is very interesting and challenging (i.e. scientific idea creation). However, this paper does not use sufficiently innovative approaches to solve it. The entire retrieval strategy is an ensemble of previously utilized techniques, such as entity matching, semantic embedding search, co-citation (this is implemented in many existing databases such as PubMed), and clustering. The idea proposal pipeline is only an implementation but with limited originality. In my opinion the most original aspect of this paper is its introduction of the annotated database where each paper has its ideas extracted and processed, though the paper does not talk about quantitative quality control of the database creation process.

4. I have some questions/doubts about the evaluation section in the paper, I will reserve them to the question section.

**Questions:**

1. I am concerned about potential data contamination. The GPT-4 model used for idea generation has a knowledge cutoff of Oct 2023, and the experiment uses ACL 2024 published articles. Surely the authors are aware that (at least some) ACL published papers went through one or many rounds of ARR rolling review, and there is no policy to prevent them at any point to publish a preprint version of their paper on the arXiv even before the paper is officially published. The question to the author is, how many papers in the experiment setting have a preprint version dated before November 2023? This must include similar versions that may went through previous rounds of ARR.

2. I don't understand why an LLM would be capable of assessing the feasibility of an idea. Especially when the criteria for feasibility in your prompt includes resource, time, and execution. Let's use the example you provided in the paper, the one about contrastive multimodal clustering. Neither the SciPIP nor ground-truth idea includes experiment plans, how many GPUs, how to collect data, the size and scale of data, and the specific formulation of the multimodal contrastive loss. How could an LLM judge idea feasibility based on almost no details? I think it would be best if this part of the evaluation be done by expert human annotators with published records in similar conferences. If resource constraints is a problem, at least verify the inter-annotator agreement rate (w.r.t. spearman correlation, cohen's kappa) between human experts and LLM judges on 100 randomly selected instances, and preferably with a hypotheses test that confirms your prompt-engineered LLM judge has performance comparable to human judges. Similar concern about novelty, it is a subjective standard, and would require verifications with human experts.

3. Going back this example you provided, as you know, the concept of multimodal contrastive learning is not new (at least already exist in 2021, https://arxiv.org/abs/2302.06232, https://arxiv.org/abs/2104.12836). Since the originality of the ground-truth lies in "construct augmentation views of multimodal data that mask either video or audio modality", why would the LLM judge give a score 4 to the SciPIP generated idea that does not describe this strategy? I examined the SciPIP generated strategy closely and it seems to be spending sentences explaining what is multimodal contrastive learning ("that optimize cosine similarity for positive pairs.."), this strategy seems more pedantic than original because the concept of multimodal contrastive learning already exists, and contrastive learning is a very popular strategy for embedding and representation learning (since SimCSE). I would expect the LLM idea proposer to at least come up with the mathematical formulation for the loss function and the plan to use it accordingly.

4. As mentioned above, please consider including more baseline models for the scientific literature retrieval experiment, and using more standard IR metric such as NDCG, Precision, not just Recall. You should also consider testing your retrieval strategy on existing Scientific Document Information retrieval benchmarks such as DORIS-MAE, SciDocs, SciRepEval etc.

5. You should discuss the applicability and generalizability of this idea proposer pipeline in other domains, such as physics, mathematics and evidence-based biomedicine. I think the prompt engineering specialization on your pipeline is tailored for AI/ML/NLP, and might not be suitable for other domains, where explicit biomedical experiments, or mathematical formulations are the core of new ideas. Otherwise, please acknowledge that this pipeline is limited to NLP research at this point.

---

### Note · Authors · 2024-11-25

I have read and agree with the venue's withdrawal policy on behalf of myself and my co-authors.